# VisCo Grids: Surface Reconstruction with Viscosity and Coarea Grids

**Albert Pumarola**[*1], **Artsiom Sanakoyeu**[*1], **Lior Yariv**[2], **Ali Thabet**[1], **Yaron Lipman**[1,2]

[1]Meta AI, [2]Weizmann Institute of Science

## Abstract

Surface reconstruction has been seeing a lot of progress lately by utilizing Implicit Neural Representations (INRs). Despite their success, INRs often introduce hard to control inductive bias (i.e., the solution surface can exhibit unexplainable behaviours), have costly inference, and are slow to train. The goal of this work is to show that replacing neural networks with simple grid functions, along with two novel geometric priors achieve comparable results to INRs, with instant inference, and improved training times. To that end we introduce VisCo Grids: a grid-based surface reconstruction method incorporating Viscosity and Coarea priors. Intuitively, the Viscosity prior replaces the smoothness inductive bias of INRs, while the Coarea favors a minimal area solution. Experimenting with VisCo Grids on a standard reconstruction baseline provided comparable results to the best performing INRs on this dataset.

## 1 Introduction

Reconstructing 3D surfaces from sparse point clouds is a long standing problem in both computer vision and graphics [7]. Methods tackling this problem aim to estimate 3D surfaces given as input unordered point sets (point clouds), with or without corresponding normals. Surface representations can be divided to two groups: parametric and implicit. Parametric methods represents the surface using some parametric domain, while implicit methods represent the surface as some level-set, $\mathcal{S} = \{p \in \mathbb{R}^3 | f(p) = c\}$, of a volumetric function $f : \mathbb{R}^3 \to \mathbb{R}$. While parametric methods can easily sample the surface, implicit methods can readily adapt

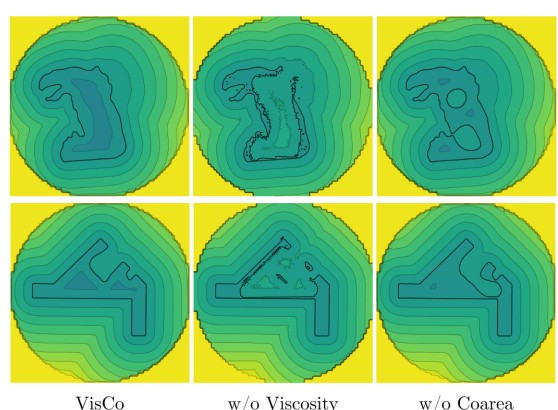

VisCo     w/o Viscosity     w/o Coarea

Figure 1: The VisCo prior (left) is incorporating viscosity and Coarea; Middle and right shows ablations on each.

to topological changes of the reconstructed surface. Parametric methods include, e.g., meshes and spline surfaces, while implicit methods use, e.g., volumetric data structures such as voxel grids, Radial Basis Functions (RBFs), or (recently) neural networks.

Implicit Neural Representation (INR) [29, 35, 13, 3, 16, 18, 4, 37] is categorized as an implicit method using a neural networks to define the implicit function $f$. INRs build upon the inherit inductive bias in neural networks and their optimization process to provide smooth yet flexible and expressive

---

[*]These authors contributed equally to this work.

36th Conference on Neural Information Processing Systems (NeurIPS 2022).

surface reconstructions. INRs have several disadvantages: First, the neural inductive bias is hard to control, often introducing undesired or unexplained surface behaviours. In fact, a considerable research effort is dedicated to fix/change/control this bias [45, 43, 27, 26]. Second, INRs have an increased deployment cost, requiring many network evaluations for surface contouring, e.g., with marching cube based methods [29, 35], or direct rendering [49, 31]. Lastly, although using high optimized solvers, INRs are still slow to train.

The goal of this work it show that network-free grid-based implicit representations can achieve INR-level reconstructions when incorporating suitable priors. To that end, we present VisCo Grids: a grid-based surface reconstruction algorithm that incorporates well-defined geometric priors: Viscosity and Coarea. In short, VisCo, see Figure 1, right. The viscosity loss, is replacing the Eikonal loss [18, 43] used in INRs for optimizing Signed Distance Functions (SDF). The Eikonal loss posses many bad minimal solution that are avoided in the INR setting due to the network's inductive bias, but are present in the grid parametrization, see e.g., Figure 1, middle. The viscosity loss, uses the notion of vanishing viscosity to regularize the Eikonal loss and provide well defined smooth solution that converges to the "correct" viscosity SDF solution. The viscosity loss provides smooth SDF solution but do not punish excessive or "ghost" surface parts, see e.g., Figure 1 (right). Therefore, a second useful prior is the coarea loss, directly controlling the surface's area, and encourages it to be smaller. The coarea loss is defined using a "squashing" function applied to the viscosity SDF making it approximately an indicator function, and then integrates its gradient norm over the domain. Integrating the gradient norm of a function is called the Total Variation loss [12, 27] and is measuring the perimeter of indicator functions, which in our case approximates $\text{area}(\mathcal{S})$. VisCo grids (as other grid methods) have instant inference, and even with our current rather naive implementation are faster to train than INRs. Considerable training time improvement are expected with a more efficient implementation.

We tested VisCo Grids on a standard 3D reconstruction dataset, and achieved comparable accuracy to the state-of-the-art INR methods. Through ablations, we demonstrate the properties and benefit in the VisCo prior.

## 2   Related Work

**3D Surface Reconstruction**   Classical approaches for surface reconstruction from point clouds [7] are either parametric [2] or implicit with mostly linear function bases, e.g., grids or radial basis functions [10, 24]. Recent works have developed methods for surface reconstruction using neural networks, which consist of a non-linear function space, making these methods non-convex. Those methods differ by how they choose to represent the 3D reconstruction. [20, 46, 21] employ a parametric point of view. Such discretizations do not yield watertight reconstruction, and/or lack topological detail. A more flexible solution is the Implicit Neural Representation (INR). INRs based methods [35, 29, 3, 13, 18, 43, 27, 6] show great progress in leveraging the inductive bias of MLPs to represent smooth surfaces, using additional losses and regularizers. For example, [27] introduce a perturbed Dirichlet loss (i.e., norm of gradient) to push for a unique and regular occupancy solution; [6] incorporates a Divergence loss (i.e., absolute value of the divergence of the gradient of trained distance field) for encouraging the learned field to resemble a gradient field of a true distance functions. Neural Spline [47] does not use neural networks directly, rather derive a kernel-based formulation arising from infinitely-wide shallow networks. Shape As points (SAP) [37] represent the surface using a differentiable Poisson solver and contouring process.

**Grid-based representations**   Recent works suggested to reduce, completely or partially, the use of neural networks in implicit representations and replacing it with a grid-based data structure. This is due to the heavy computational resources required in optimizing and evaluating neural networks. Plenoxels [1] propose a view-dependent sparse voxel model and show comparable results to NeRF [31] and a speedup of two orders of magnitude. Neural Geometric Level of Detail [44] uses an octree-based feature volume and a small MLP to represent SDF. [32] shows fast training of INR's using a small neural network augmented by a multiresolution hash table with trainable features. Similar to DeepSDF [35], both work used 3D supervision for learning the SDF.

# 3 Method

We consider the 3D euclidean space $\mathbb{R}^3$ with points $p = (x, y, z) \in \mathbb{R}^3$. We discretize the unit cube $\mathcal{C} = [0, 1]^3$ with a 3D voxel grid $\mathcal{G} = \{p_I\}$, with nodes $p_I$ indexed by $I = (i, j, k)$, $i, j, k \in [n] = \{1, \ldots, n\}$, i.e., $p_I = (x_{ijk}, y_{ijk}, z_{ijk})$. We denote by $h = n^{-1}$, and by $N = n^3$ the total number of nodes. We represent our reconstructed surface as a zero level of a scalar function $f$ defined over the cube $\mathcal{C}$. $f$ is defined by prescribing its values at the grid's nodes $f_I \in \mathbb{R}$ and trilinear interpolating in each voxel. We will denote by $f(p)$ the interpolated value at point $p$.

Given an input point cloud consisting of $m$ points $q_k \in \mathbb{R}^3$ with or without (unit norm) normals $n_k \in \mathbb{R}^3$, $k \in [m]$, our goal is to compute $f$ so that its zero level set approximates the unknown surface, i.e.,

$$\mathcal{S} = \{p \in \mathcal{C} \mid f(p) = 0\}. \tag{1}$$

Our approach to compute $f$ is to minimize a loss function of the form

$$\mathcal{L} = \mathcal{L}_{\text{data}} + \mathcal{L}_{\text{prior}} \tag{2}$$

where

$$\mathcal{L}_{\text{data}} = \frac{\lambda_{\text{p}}}{m} \sum_{k=1}^{m} |f(q_k)|^2 + \frac{\lambda_{\text{n}}}{m} \sum_{k=1}^{m} \|\nabla f(q_k) - n_k\|^2 \tag{3}$$

where $\|\cdot\|$ is the standard euclidean norm in $\mathbb{R}^3$, $\nabla f(p) \in \mathbb{R}^3$ is the gradient of $f$ sampled at point $p$. Note that $\nabla f$ is defined in interior of voxels, which is generically where the input points $q_k$ resides. $\mathcal{L}_{\text{data}}$ is the standard data loss encouraging the zero level to pass through the input points $q_k$, and its normals (defined by gradients of $f$) to coincide with input normals $n_k$.

The prior, $\mathcal{L}_{\text{prior}}$, is the main contribution of this work, where we combine two novel losses,

$$\mathcal{L}_{\text{prior}} = \lambda_{\text{v}} \mathcal{L}_{\text{viscosity}} + \lambda_{\text{c}} \mathcal{L}_{\text{coarea}} \tag{4}$$

Intuitively, the viscosity loss optimizes for a smooth Signed Distance Function (SDF) solutions, avoiding auxiliary bad minima of the Eikonal equation, while the coarea loss strives to minimize the area of the zero level surface. Our loss has 4 hyper-parameters $\lambda_{\text{p}}, \lambda_{\text{n}}, \lambda_{\text{v}}, \lambda_{\text{c}}$. We provide more details on these priors next.

## 3.1 Viscosity Loss

The goal of the viscosity loss is to make $f$ approximate an SDF over $\mathcal{C}$. Given boundary conditions asking $f$ to vanish on some closed compact surface $\mathcal{S}$, the SDF solves the Eikonal equation PDE, i.e., $\|\nabla f(p)\| = 1$, in a certain well defined sense (viscosity). This motivated some previous work to directly optimize the Eikonal loss [18, 43]

$$\mathcal{L}_{\text{eikonal}} = \int_{\mathcal{C}} \left( \|\nabla f(p)\| - 1 \right)^2 dp \tag{5}$$

Unfortunately, the Eikonal loss has many undesirable minima which are not SDFs. Figure 2 shows a 1D example: both depicted solutions (denoted $f$) vanish at the input points $q_1, q_2$ (black points) and globally minimize the Eikonal loss over the grid (grid points are shown in blue). The INR works mentioned above use neural networks for representing $f$ which injects an inductive bias avoiding these bad minima, however on grids, minimizing equation 5 cannot avoid these solutions. See, e.g., middle column in Figure 1.

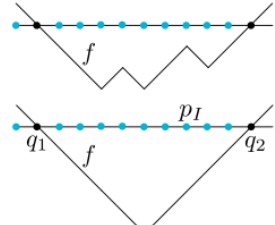

More classical Eikonal solvers do work with grids however use mostly fast marching or sweeping methods [33, 41, 50, 11]. Namely, use a special discretization of the Eikonal equation favoring the viscosity solution of the Eikonal [40], and update node values according to a moving front [41]. Since this discretization is up-wind (will only propagate values in one direction) and requires choosing the maximal among its solution, its success in adaptation to a loss is not clear.

Figure 2: Two global minimizers of the Eikonal loss over a grid in 1D. Top solution is not an SDF.

We use a different approach to build a loss favoring SDF solutions over grids motivated by the vanishing viscosity method [15]. Namely, adding to the Eikonal PDE a small perturbation of the

Laplacian of $f$ (denoted by $\Delta f$), i.e., $\|\nabla f(p)\| - 1 - \epsilon \Delta f(p) = 0$, makes the PDE semi-linear elliptic [9], and hence with suitable boundary conditions it is uniquely solvable inside $\mathcal{S}$ with a smooth solution, approaching the viscosity positive distance function to the boundary as $\epsilon \to 0$. Similarly, for $1 - \|\nabla f(p)\| - \epsilon \Delta f(p) = 0$ the solution approaches the negative distance function inside the domain. Motivated by the vanishing viscosity principle we suggest the following viscosity loss:

$$\mathcal{L}_{\text{viscosity}} = \int_{\mathcal{C}} \Big( (\|\nabla f(p)\| - 1)\text{sign}(f(p)) - \epsilon \Delta f(p) \Big)^2 dp \tag{6}$$

We discretize this loss over the grid $\mathcal{G}$ by replacing the first order derivatives and second order derivatives with symmetric finite differences, i.e.,

$$D_x f_I = D_x f_{i,j,k} = \frac{f_{i+1,j,k} - f_{i-1,j,k}}{2h}, \quad D_x^2 f_I = D_x^2 f_{i,j,k} = \frac{f_{i+1,j,k} - 2f_{i,j,k} + f_{i-1,j,k}}{h^2}$$

and similarly for $D_y$ and $D_z$. We use these discrete operators to approximate the gradient $\widehat{\nabla} f(p_I) = (D_x f_I, D_y f_I, D_z f_I)$ and Laplacian $\widehat{\Delta} f(p_I) = D_x^2 f_I + D_y^2 f_I + D_z^2 f_I$. The discretized viscosity loss now takes the form

$$\widehat{\mathcal{L}}_{\text{viscosity}} = \frac{1}{N} \sum_I \Big( (\|\widehat{\nabla} f(p_I)\| - 1)\text{sign}(f(p_I)) - \epsilon \widehat{\Delta} f(p_I) \Big)^2 \tag{7}$$

### 3.2 Coarea loss

The coarea loss is approximating the area of the zero level set, and therefore incorporating it in the optimization pushes the reconstructed surface to be economic in area.

First, similarly to [48] we use the centered Laplace CDF

$$\Psi_\beta(s) = \begin{cases} \frac{1}{2} \exp\left(\frac{s}{\beta}\right) & s \leq 0 \\ 1 - \frac{1}{2} \exp\left(-\frac{s}{\beta}\right) & s \geq 0 \end{cases} \tag{8}$$

to transform the SDF $f$ to a smooth approximation of the indicator function:

$$\chi_\beta(p) = \Psi_\beta(-f(p)) \tag{9}$$

As $\beta \to 0$, $\chi_\beta$ converges to an indicator function leading to 1 inside $\mathcal{S}$ and 0 outside. The coarea loss is defined as

$$\mathcal{L}_{\text{coarea}} = \int_{\mathcal{C}} \|\nabla \chi_\beta(p)\| \, dp \tag{10}$$

To understand why this loss approximates the area of $\mathcal{S}$ we can use the coarea formula [39]:

$$\int_{\mathcal{C}} \|\nabla \chi_\beta(p)\| \, dp = \int_{-\infty}^{\infty} \text{area}(\chi_\beta^{-1}(s)) ds, \tag{11}$$

where $\chi_\beta^{-1}(s) = \{p \mid \chi_\beta(p) = s\}$ is the preimage of the value $s$. Since $\chi_x(p) \in [0, 1]$ the r.h.s. integral can be restricted to the interval $[0, 1]$, and therefore the coarea loss averages the area of the level sets of $\chi_\beta$. Next,

$$\chi_\beta^{-1}(s) = \{p \mid \Psi_\beta(-f(p)) = s\} = \{p \mid f(p) = -\Psi_\beta^{-1}(s)\} = f^{-1}(-\Psi_\beta^{-1}(s)),$$



which shows that the level set $s \in (0, 1)$ of $\chi_\beta$ is the level set $-\Psi_\beta^{-1}(s)$ of the SDF $f$. As $\beta \to 0$, $-\Psi_\beta^{-1}(s) \to 0$ for all $s \in (0, 1)$ (and uniformly in $(\epsilon, 1 - \epsilon)$ for fixed $\epsilon > 0$). Therefore the average of the level set area (i.e., the r.h.s. of equation 11) converges to the area of $f^{-1}(0) = \mathcal{S}$. Figure 1 (right) shows how removing the coarea loss introduces an extraneous zero level set, and hence results in an undesired surface part. Figure 3 shows a comparison of a reconstruction of semisphere with and without coarea. In the experiments section we provide more ablation tests with the coarea and viscosity losses.

Figure 3: Reconstruction of a semisphere point cloud (white dots) without (left) and with (right) coarea loss.

To discretize the coarea loss we let $w_I$ denote the centers of grid's voxels, and note that $\nabla \chi_\beta(w_I) = \Phi_\beta(-f(w_I))\nabla f(w_I)$, where

$$\Phi_\beta(s) = \frac{1}{2\beta} \exp\left(\frac{|s|}{\beta}\right)$$

is the PDF of the Laplace distribution, and $\nabla f(w_I)$ is computed as a linear combination of the voxel's corner values $f_{I_1}, \ldots, f_{I_8}$, see more details in the Appendix. We end up with the discretized loss:

$$\widehat{\mathcal{L}}_{\text{coarea}} = \frac{1}{N} \sum_I \Phi_\beta(-f(w_I)) \left\| \nabla f(w_I) \right\| \qquad (12)$$

This loss is usually incorporated with a small hyper-parameter $\lambda_c$ with the purpose of eliminating redundant surface parts.

## 4 Experiments

In this section we extensively evaluate VisCo grids. First, we evaluate on two standard surface reconstruction benchmarks [46, 22] (Sec. 4.1) against a large variety of state-of-the-art methods: Poisson Surface Reconstruction [24], DGP [46], IGR [18], SIREN [43], FFN [45], NSP [47], PHASE [27], GD [14], BPA [8], SPSR [25], RIMLS [34], SALD [5], IGR [19], OccNet [30], DeepSDF [36], LIG [23], Points2Surf [17], DSE [38], IMLSNet [28] and ParseNet [42]. We then perform an ablation study (Sec. 4.2), and conduct a detail examination of the main components of the model, namely the viscosity and coarea losses. Finally, we discuss the model's ability to reconstruct sparse point clouds (Sec. 4.3) using scans from Stanford 3D Scanning Repository.

### 4.1 Surface reconstruction benchmarks

We next evaluated our model on two benchmarks: Surface Reconstruction Benchmark [46] and Surface Reconstruction from Real-Scans [22]. Each containing challenging object with complex shape and topology. Importantly, we use same hyper-paramenters for all meshes of all benchmarks with no extensive hyper-parameter search.

**Surface Reconstruction Benchmark** This benchmark [46] consists of 5 noisy range scans, each containing point cloud and normal data. We evaluate our method against current state of the art methods on this benchmark: Deep Geometric Prior (DGP) [46], Implicit Geometric Regularization (IGR) [18], SIREN [43], Fourier Feature Networks (FFN) [45], NSP [47] and PHASE [27]. We additionally compare to the classical method of Poisson Surface Reconstruction [24]. Quantitative results are summarized in Table 1. We report the Chamfer ($d_C$) and Hausdorff ($d_H$) distances between the reconstructed meshes and the ground-truth point clouds. Furthermore, we report their corresponding one sided distances ($\vec{d_H}$ and $\vec{d_C}$) between the reconstructed meshes and the input noisy point cloud. Representative qualitative results are shown in Figure 5. Note that we achieve comparable results to the current state-of-the-art INR methods.

**Surface Reconstruction from Real-Scans** This benchmark [22] consists of 21 noisy range scans of real objects. We evaluate our method against: GD [14], BPA [8], SPSR [25], RIMLS [34], SALD [5], IGR [19], OccNet [30], DeepSDF [36], LIG [23], Points2Surf [17], DSE [38], IMLSNet [28] and ParseNet [42]. Quantitative results are summarized in Table 2. We report Chamfer Distance ($d_C$), F-score, Normal Consistency Score (NCS) [30], and Neural Feature Similarity (NFS) [22] distances between the reconstructed meshes and the ground-truth point clouds. Furthermore, we report the one sided distances ($\vec{d_H}$ and $\vec{d_C}$) between the reconstructed meshes and the input noisy point cloud. Representative qualitative results are shown in Figure 5. Note that we achieve 1-st to 3-rd place across all categories, with top F-score.

### 4.2 Ablation study

We provide an ablation study of the main components of our model in Table 3 and Figure 6. Specifically we compared with the following alternatives: i) Eikonal loss without the viscosity term that prevents undesirable non-SDF solutions, i.e., $\epsilon = 0$ in equation 6; ii) removing the coarea loss enforcing minimal surface area, i.e., $\lambda_c = 0$; and iii) removing the normal loss, i.e., $\lambda_n = 0$. Note

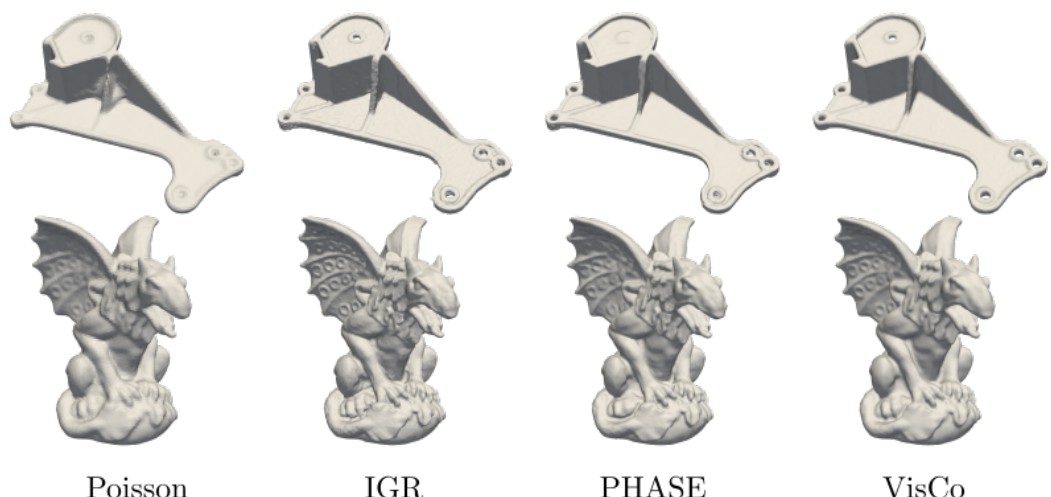

Figure 4: Qualitative results for surface reconstruction [46] compared to existing methods. Note how VisCo Grids achieve comparable level of details when compared to other baselines.

| | | Poisson | DGP | IGR | SIREN | FFN | NSP | PHASE | **Ours** (30 mins) | **Ours** (8 mins) |
|---|---|---|---|---|---|---|---|---|---|---|
| Anchor | $d_C$ | 0.60 | 0.33 | 0.22 | 0.32 | 0.31 | 0.22 | **0.21** | **0.21** | 0.28 |
| | $d_H$ | 14.89 | 8.82 | 4.71 | 8.19 | 4.49 | 4.65 | 4.29 | **3.00** | 5.69 |
| | $d_{\vec{C}}$ | 0.60 | 0.08 | 0.12 | 0.10 | 0.10 | 0.11 | 0.09 | 0.15 | 0.15 |
| | $d_{\vec{H}}$ | 14.89 | 2.79 | 1.32 | 2.43 | 0.10 | 1.11 | 1.23 | 1.07 | 1.15 |
| Daratech | $d_C$ | 0.44 | 0.20 | 0.25 | 0.21 | 0.34 | 0.21 | **0.18** | 0.26 | 0.25 |
| | $d_H$ | 7.24 | 3.14 | 4.01 | 4.30 | 5.97 | 4.35 | **2.92** | 4.06 | 4.15 |
| | $d_{\vec{C}}$ | 0.44 | 0.04 | 0.08 | 0.09 | 0.10 | 0.08 | 0.08 | 0.14 | 0.13 |
| | $d_{\vec{H}}$ | 7.24 | 1.89 | 1.59 | 1.77 | 0.10 | 1.14 | 1.80 | 1.76 | 1.78 |
| DC | $d_C$ | 0.27 | 0.18 | 0.17 | 0.15 | 0.20 | **0.14** | 0.15 | 0.15 | 0.15 |
| | $d_H$ | 3.10 | 4.31 | 2.22 | 2.18 | 2.87 | **1.35** | 2.52 | 2.22 | 2.23 |
| | $d_{\vec{C}}$ | 0.27 | 0.04 | 0.09 | 0.06 | 0.10 | 0.06 | 0.05 | 0.09 | 0.09 |
| | $d_{\vec{H}}$ | 3.10 | 2.53 | 2.61 | 2.76 | 0.12 | 2.75 | 2.78 | 2.76 | 2.78 |
| Gargoyle | $d_C$ | 0.26 | 0.21 | **0.16** | 0.17 | 0.22 | **0.16** | **0.16** | 0.17 | 0.17 |
| | $d_H$ | 6.8 | 5.98 | 3.52 | 4.64 | 5.04 | 3.20 | **3.14** | 4.40 | 4.45 |
| | $d_{\vec{C}}$ | 0.26 | 0.06 | 0.06 | 0.08 | 0.09 | 0.08 | 0.07 | 0.11 | 0.11 |
| | $d_{\vec{H}}$ | 6.80 | 3.41 | 0.81 | 0.91 | 0.09 | 2.75 | 1.09 | 0.96 | 0.98 |
| Lord Quas | $d_C$ | 0.20 | 0.14 | 0.12 | 0.17 | 0.35 | 0.12 | **0.11** | 0.12 | 0.13 |
| | $d_H$ | 4.61 | 3.67 | 1.17 | 0.82 | 3.90 | **0.69** | 0.96 | 1.06 | 1.14 |
| | $d_{\vec{C}}$ | 0.20 | 0.04 | 0.07 | 0.12 | 0.06 | 0.05 | 0.04 | 0.07 | 0.07 |
| | $d_{\vec{H}}$ | 4.61 | 2.03 | 0.98 | 0.76 | 0.06 | 0.62 | 0.96 | 0.64 | 0.68 |

Table 1: Surface reconstruction results on the benchmark of [46]. We show reconstruction results for each model for our method at 256 grid resolution with 30 minute and 8 minute time budget. We also show results from comparative methods. Bold numbers signify top performance. We report Chamfer and Hausdorff distances using ground truth scans ($d_C$, $d_H$) and input scans ($d_{\vec{C}}$, $d_{\vec{H}}$). Note that VisCo Grids achieve comparable results to SOTA INRs, and even matches it in terms of Chamfer distance in 3 out of 5 meshes.

that without coarea and viscosity the reconstruction tends to have holes and discontinuities near the surface boundaries. Only combination of all the components results in a good surface reconstruction.

**Learning with viscosity.** We further provide a more in depth discussion of the proposed viscosity loss. Figure 7 compares reconstructions with different levels of the viscosity parameter, i.e., $\epsilon$ in equation 6. As can be inspected from this figure, viscosity affects the smoothness of the reconstructed surface. For a low viscosity parameter the zero level sets become noisy. This can be explained by the viscosity eikonal loss (i.e., equation 6) becoming numerically too close to the eikonal loss in

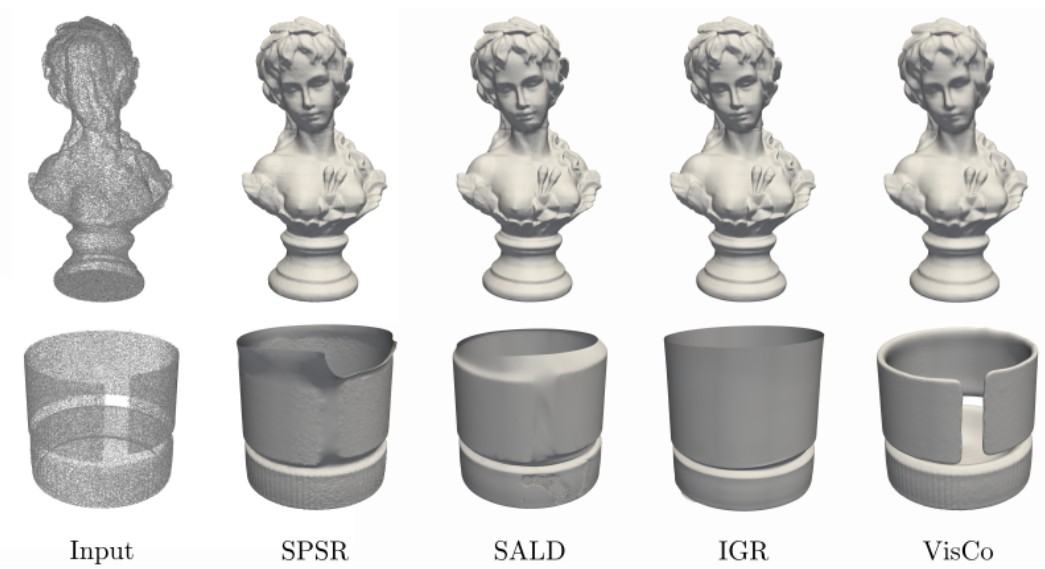

| Input | SPSR | SALD | IGR | VisCo |

Figure 5: Qualitative results for surface reconstruction of real objects [22] compared to existing methods. Note how VisCo Grids does not over-extend the surface in the bottom row example. The competing methods meshes were provided by the benchmark organizers.

| Prior | Method | $d_C$ $(\times 10^{-2})\downarrow$ | F-score (%) ↑ | NCS $(\times 10^{-2})$ ↑ | NFS $(\times 10^{-2})$ ↑ |
|---|---|---|---|---|---|
| Triangulation-based | GD [14] | 31.72 | 87.51 | 88.86 | 82.20 |
| | BPA [8] | 40.37 | 80.95 | 87.56 | 68.69 |
| Smoothness | SPSR [25] | **31.05** | 87.74 | 94.94 | **89.38** |
| | RIMLS [34] | 32.80 | 87.05 | 91.97 | 85.19 |
| | **Ours** | 32.11 ($3^{rd}$) | **88.52** ($1^{st}$) | 94.20 ($3^{rd}$) | 89.16 ($2^{rd}$) |
| Modeling | SALD [5] | 31.13 | 87.72 | 94.68 | 86.86 |
| | IGR [19] | 32.70 | 87.18 | **95.99** | 89.10 |
| Learning Semantics | OccNet [30] | 232.71 | 17.11 | 80.96 | 39.70 |
| | DeepSDF [36] | 263.92 | 19.83 | 77.95 | 40.95 |
| Local Learning | LIG [23] | 48.75 | 83.76 | 92.57 | 81.48 |
| | Points2Surf [17] | 48.93 | 80.89 | 89.52 | 81.83 |
| Hybird | DSE [38] | 32.16 | 86.88 | 87.20 | 76.81 |
| | IMLSNet [28] | 38.46 | 82.44 | 93.31 | 85.30 |
| | ParseNet [42] | 149.96 | 38.92 | 81.51 | 45.67 |

Table 2: Surface reconstruction results on the 20 real-scanned benchmark [22] meshes. We report Chamfer Distance ($d_C$), F-score, Normal Consistency Score (NCS) [30], and Neural Feature Similarity (NFS) [22]. Methods are grouped according to surface geometry priors, as originally defined in the benchmark. Our method achieves top F-score and 1-st to 3-rd place in all scores.

equation 5 and the solution deviates from the viscosity SDF solution. This leads to artifacts across the surface, similar to the limit case ($\epsilon = 0$) where only the eikonal loss is used, see the second column from the left. For a high viscosity parameter, and as expected with the addition of a non-vanishing Laplacian term, the surface becomes over-smoothed.

**Learning with coarea.** Similarly, we also provide an analysis of the proposed coarea loss. In Figure 8 we show the effect of changing the parameter weight of the coarea loss, $\lambda_c$. As can be observed in the figure, a low coarea weight leads to the presence of excessive surface area in the reconstruction. In contrast, a high weight will strive to minimize the surface area. In a sense, the coarea serves a surface tension parameter; stronger tension will ignore points, weaker tension will overfit.

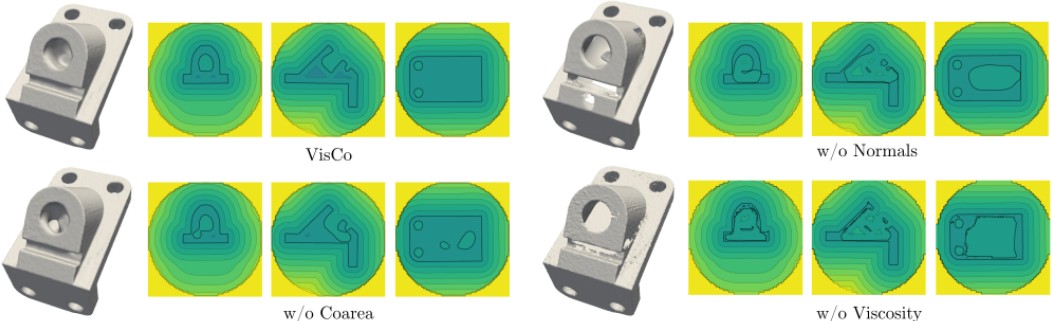

Figure 6: Ablation for the main components of our method. Removing elements of our loss leads to subpar reconstructions. We can observe these artifacts in the level sets shown in this figure. Removing viscosity results in discontinuities in the final surface, while no coarea produces excess surface area.

|  |  | Baseline | w/o normals | w/o viscosity | w/o coarea |
|---|---|---|---|---|---|
| Anchor | $d_C$ | **0.21** | 0.61 | 0.55 | 0.72 |
|  | $d_H$ | **3.00** | 7.82 | 10.83 | 10.24 |
|  | $d_C^{\rightarrow}$ | 0.15 | 0.37 | 0.27 | 0.36 |
|  | $d_H^{\rightarrow}$ | 1.07 | 7.84 | 1.44 | 9.68 |
| Daratech | $d_C$ | 0.26 | 0.24 | 0.24 | **0.23** |
|  | $d_H$ | 4.06 | 4.2 | 4.3 | **2.19** |
|  | $d_C^{\rightarrow}$ | 0.14 | 0.13 | 0.12 | 0.13 |
|  | $d_H^{\rightarrow}$ | 1.76 | 2.69 | 1.77 | 1.77 |
| DC | $d_C$ | **0.15** | **0.15** | **0.15** | 0.34 |
|  | $d_H$ | **2.22** | 2.24 | 2.24 | 6.58 |
|  | $d_C^{\rightarrow}$ | 0.09 | 0.08 | 0.08 | 0.16 |
|  | $d_H^{\rightarrow}$ | 2.76 | 2.76 | 2.79 | 2.82 |
| Gargoyle | $d_C$ | **0.17** | 0.58 | 0.47 | 0.59 |
|  | $d_H$ | **4.40** | 6.32 | 10.38 | 6.35 |
|  | $d_C^{\rightarrow}$ | 0.11 | 0.07 | 0.26 | 0.38 |
|  | $d_H^{\rightarrow}$ | 0.96 | 2.39 | 1.34 | 1.25 |
| Lord Quas | $d_C$ | **0.12** | 0.12 | 0.12 | 0.58 |
|  | $d_H$ | 1.06 | 1.38 | **1.04** | 6.05 |
|  | $d_C^{\rightarrow}$ | 0.07 | 0.37 | 0.06 | 0.32 |
|  | $d_H^{\rightarrow}$ | 0.64 | 0.69 | 0.64 | 3.73 |

Table 3: Ablations study. We show the contribution of each component of VisCo Grids. Baseline is the full method. The remaining columns correspond to optimizing without normal loss, viscosity loss and coarea loss, respectively. We show results for each mesh of the benchmark [46]. The results justify the use of the different components in VisCo Grids.

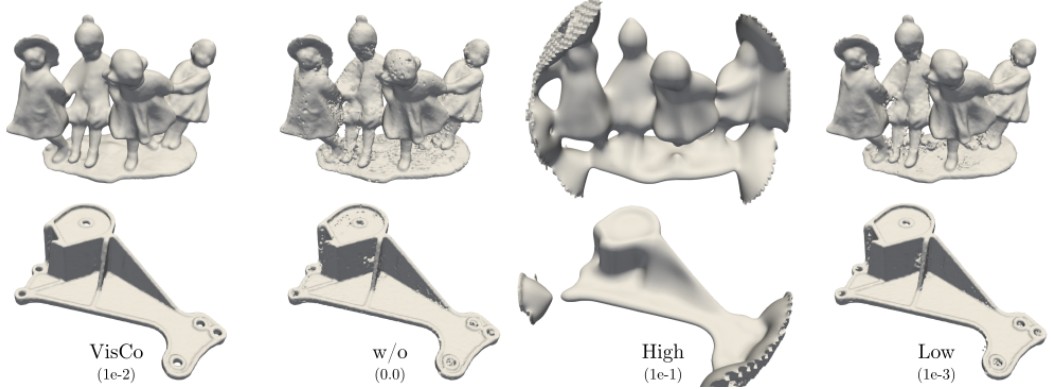

Figure 7: Viscosity loss ablation. Setting a high viscosity loss parameter, $\epsilon$, leads to oversmoothing. In contrast, setting it too low leads to noise and discontinuities in the surfaces, similarly to removing it by setting $\epsilon = 0$.

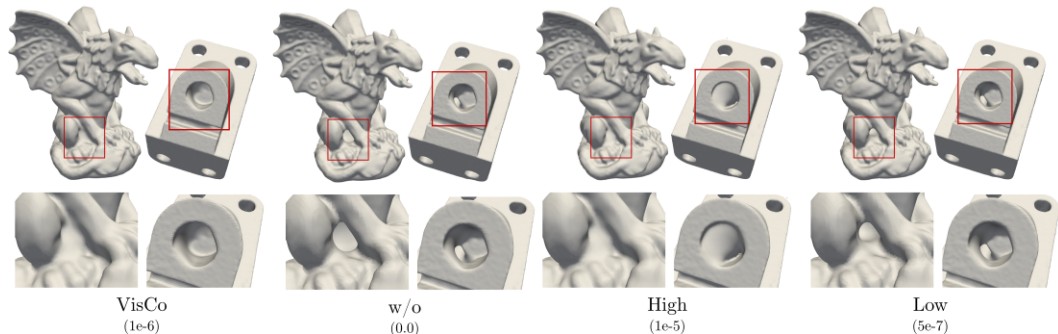

VisCo (1e-6)     w/o (0.0)     High (1e-5)     Low (5e-7)

Figure 8: Coarea loss ablation. Coarea loss favors solution with low surface area. Here, note how larger coarea weight tends to fill in the cavities and close the gaps in the shape. In contrast, very low weight fails to drive the optimization procedure towards a solution with smaller surface area.

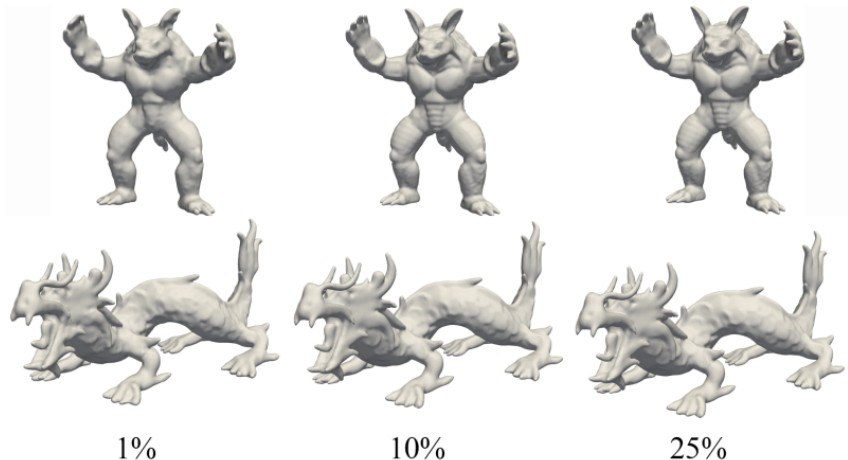

1%        10%        25%

Figure 9: VisCo reconstructions from sparse point cloud inputs.

### 4.3 Reconstructing from sparse point clouds

We now evaluate our model ability to reconstruct surfaces in the challenging case of sparse input point clouds. For this experiment we use point clouds from the Stanford 3D Scanning Repository and downsample them at different levels: 1%, 10%, and 25%. In Figure 9 we visualize the VisCo reconstructions. Note that VisCo can reconstruct the shapes even with sparse input. This provides a further validation for the proposed geometrical priors.

## 5 Conclusions

We introduced VisCo Grids, a novel grid-based surface reconstruction approach that leverages two novel geometrically motivated priors: viscosity and coarea. We advocate VisCo's prior for grid functions as an alternative to the implicit inductive bias of implicit neural representations for the task of surface reconstruction. One important limitation of our method, shared by all grid methods, is that its degrees of freedom, namely nodes' location, are set a-priori. In contrast, using non-linear function spaces, such as neural networks, allows a more flexible usage of the degrees of freedoms in the model and can adjust to areas with more detail. Nevertheless, we still believe that grid functions and direct priors, incorporated with modern computing power, are valuable add-ons to the surface reconstruction toolbox, providing few clear benefits over neural networks, such as a well-understood control over surface properties, instant inference time, and faster training.

**Acknowledgements.** We thank the anonymous reviewers for their constructive comments. LY was supported by a grant from Israel CHE Program for Data Science Research Centers.

**Social impact.** We don't see any immediate negative societal impact from our work. However, we acknowledge that high quality 3D reconstruction in general can be used in malicious settings.

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
