# OpenReview forum: "VisCo Grids: Surface Reconstruction with Viscosity and Coarea Grids"
_NeurIPS.cc/2022/Conference — NeurIPS 2022 Accept_

### Official Review · Reviewer_ubX9 · 2022-07-10

**Rating:** 4
**Confidence:** 4
**Soundness:** 3 good
**Presentation:** 3 good
**Contribution:** 2 fair

**Summary:**

The proposed method tries to improve the surface reconstruction quality by introducing two losses.
The viscosity loss tries to imporve the smoothness of the surface part without points.
The coarea loss tries to approximate the area of the zero level set.
The experiments show the proposed method can get better surface reconstruction results, especially with sparse point inputs.

**Questions:**

1. Could the authors explain why the coarea loss is introduced more clearly?
2. Could the authors do some experiments on point clouds by MVS, which have more noises than the data used in the paper?

**Limitations:**

The authors discuss the limitation in the conclusion briefly. The work does not have potential negative societal impact.

**Strengths And Weaknesses:**

Strengths
1. The paper is generally well written. All the figures and equations are clear.
2. The authors clearly explain why the viscosity loss should be introduced.
3. The experiments results on sparse point inputs are improved.

Weaknesses
1. The main contribution of the paper is the introduction of two losses. There is no neural network sturctue contribution.
2. The motivation of coarea loss is not explained clearly. We can only know its effectness by the ablation study.
3. The quantitive results on benchmarks and of the ablation study are not improved significantly. On several data of the benchmark, the top results are not generated by the proposed method. In the ablation study, the baseline method cannot beat other settings.
4. The experiments are performed on point clouds without many noises.

---

> ### Author Response · Authors · 2022-08-02
> **Answer**
>
> We thank the reviewer for his detailed and thoughtful review. Below we address the reviewer’s questions and concerns.
>
> Q1: The main contribution of the paper is the introduction of two losses. There is no neural network structure contribution.
>
> A1: The main goal of our work is to show that current neural networks representations can be *replaced* with grids and suitable losses for the task of surface reconstruction.  The benefits of grid SDF representation (over neural network SDF representation)  are mostly: (i) controllable and better understood inductive bias, (ii) fast and easy inference; and (iii) faster training times.
>
> Q2: The motivation of coarea loss is not explained clearly. We can only know its effectiveness by the ablation study. Could the authors explain why the coarea loss is introduced more clearly?
>
> A2: In Section 3.2 we provided a detailed theoretical explanation of the loss. In particular we explain how and why it approximates the zero level-set of the SDF.
>
> Q3: The quantitative results on benchmarks and of the ablation study are not improved significantly. On several data of the benchmark, the top results are not generated by the proposed method. In the ablation study, the baseline method cannot beat other settings. The experiments are performed on point clouds without many noises. Could the authors do some experiments on point clouds by MVS, which have more noises than the data used in the paper?
>
> A3: Our goal was to show that we can achieve comparable results to neural network SOTA in a fraction of the time. For example PHASE takes roughly 4 hours to optimize per model in the Surface Reconstruction Benchmark (SRB) [22], while we achieve comparable results in 8 mins.
> Furthermore, taking the reviewers’ criticism into mind, we have now added an evaluation of our method on a recently published benchmark of real scans [22] that consists of real-life scanned noise. In the revised paper, we have used the exact same parameters from the SRB dataset (i.e., no special param tuning) and got top 1-3 scores in all categories compared to a wide range of baselines; please see Tab. 2, and Fig. 5 (in the new Section “Surface Reconstruction from Real-Scans”).
>
> [22] Surface Reconstruction from Point Clouds: A Survey and a Benchmark, Zhangjin Huang, Yuxin Wen, Zihao Wang, Jinjuan Ren, Kui Jia; arXiv:2205.02413; 5 May 2022

---

### Official Review · Reviewer_WtxG · 2022-07-11

**Rating:** 6
**Confidence:** 5
**Soundness:** 3 good
**Presentation:** 4 excellent
**Contribution:** 3 good

**Summary:**

The paper proposes a method for SDF based surface reconstruction from point clouds using grid based functions, rather than the recently popular INR approaches. They use standard INR losses along with a new prior loss that has two components, a viscosity prior (for smoothness in the SDF) and a coarea prior (for minimal surface area). They show quantitative results on the SRB dataset, which are comparative to SoTA, and qualitative results on the Stanford 3D scanning repository.

**Questions:**

PHASE [18] is quite similar (a method with a direct prior based on smoothness and minimal area) and gets better (or similar) performance. Why is this the case? Is it just because of it using a neural network vs this work being network free, or could it possibily be better direct priors?

The following work might be of interest to you: https://arxiv.org/abs/2206.03087v1, it is a concurrent work (accepted to CVPR2022) with similar priors (they have them as regularizations): a Hessian regularization for smoothness and a minimal surface regularization.

**Limitations:**

Yes, limitations in conclusion, and social impact in subsection in conclusion.

**Strengths And Weaknesses:**

Strengths
- Agree that general direct priors are still important for surface reconstruction given how important the task is and how various the shapes can be, and the paper gives two interesting priors
- Mathematical explanation, intuition and visualisations of the two components of the prior all show their value
- Results with normals are comparative to SoTA
- Good ablations of the components
- Paper is well written

Weaknesses
- Limited quantitative results (5 shapes). While SRB is a well established baseline, having this as the only quantitative results is not great.
- Given that the method is not using a neural network and is grid based, a major strength of the method should be lower optimisation time, with the weakness being storage of the learned INR. A discussion of both is missing, with no comparison of optimisation time or learned storage to other methods.
- The use of minimising the Laplacian as a prior has been explored in INR work, such as DiGS [* 1] and NSP [32], and has connections to PHASE [18]. This is not discussed at all (and should be in the related work), and DiGS is not even cited.
- Poor results without normals, especially given the contribution is meant to be a good prior for SDFs and thus should give results comparable to PHASE [18] or DiGS [* 1].

[* 1] https://arxiv.org/abs/2106.10811

---

> ### Author Response · Authors · 2022-08-02
> **Answer**
>
> We thank the reviewer for his detailed and thoughtful review. Below we address the reviewer’s questions and concerns.
>
> Q1: Limited quantitative results (5 shapes). While SRB is a well established baseline, having this as the only quantitative result is not great.
>
> A1: We have now added an evaluation of our method on a recently published benchmark of real scans [22]. (As a side comment: most other existing point cloud reconstruction benchmarks lack the necessary license for open publication.)
> In the revised paper, we have used the exact same parameters from the SRB dataset (i.e., no special param tuning) and got top 1-3 results in all categories; please see Table 2, and Figure 5 (in the new Section 4.1).
>
> [22] Surface Reconstruction from Point Clouds: A Survey and a Benchmark, Zhangjin Huang, Yuxin Wen, Zihao Wang, Jinjuan Ren, Kui Jia; arXiv:2205.02413; 5 May 2022
>
> Q2: Given that the method is not using a neural network and is grid based, a major strength of the method should be lower optimisation time, with the weakness being storage of the learned INR. A discussion of both is missing, with no comparison of optimisation time or learned storage to other methods.
>
> A2: We have added a discussion of optimization times and memory usage in the revised paper’s supplementary Sec. C.
>
> We report the time and memory footprint required for a single training iteration on NVIDIA Quadro GP100 GPU. Because of the pruning applied to the grid, we need to learn only a sparse set of the grid values (we call them \emph{active}).
> * 64^3 resolution: (57% of the grid values are active): 2.3 msec, 975MB VRAM
> * 128^3 resolution:  (31% of the grid values are active): 8.6 msec, 1070MB VRAM
> * 256^3 resolution:  (30% of the grid values are active): 25.8 msec, 1650MB VRAM
>
> For neural networks (INRs) every point evaluation requires forward and backward in a network involving all network’s parameters in general, while for a grid we only require nearby grid function values (learnable parameters). Typical iteration times for NN (taken from DiGS) are:
> * 66.5K params: 5.2-12.0 msec
> * 2.1M params: 17.5 msec
>
> Q3: The use of minimising the Laplacian as a prior has been explored in INR work, such as DiGS [* 1] and NSP [48], and has connections to PHASE [28]. This is not discussed at all (and should be in the related work), and DiGS is not even cited.
>
> A3: Thank you for pointing this out. We added DiGS to the references and discussed DiGS and PHASE related regularization ideas. Please see the revised paper.
>
> Q4: PHASE [28] is quite similar (a method with a direct prior based on smoothness and minimal area) and gets better (or similar) performance. Why is this the case? Is it just because of it using a neural network vs this work being network free, or could it possibly be better direct priors?
>
> A4: PHASE is neural network based that learns a smooth occupancy function where the Signed Distance Function (SDF) is computed via a log transform. For the Surface Reconstruction Benchmark (SRB) [47], PHASE takes about 4 hours to train over a single example. In our paper we show that one can work directly with SDF over grids by incorporating the Viscosity loss, and use a CoArea loss to directly control level-set area of the SDF. This allows us to achieve comparable results to PHASE in a fraction of the time (8 mins). As for comparing the priors: it is difficult to factor out the neural network bias on PHASE. We can say that the Coarea loss has a more direct influence on the zero level-set area, and the Viscosity loss encourages the solution to converge to the SDF of the zero crossings.

---

> > ### Comment · Reviewer_WtxG · 2022-08-09
> > **Reply to the Author's Response**
> >
> > Thank you for the response. My questions have been sufficiently addressed and some of my feedback has been integrated into the revised paper.
> >
> > I have also read the other reviews and the authors' response to those papers, and have no further questions.
> >
> > I stand with my original rating and recommend accepting the paper.

---

### Official Review · Reviewer_tYGS · 2022-07-11

**Rating:** 4
**Confidence:** 4
**Soundness:** 3 good
**Presentation:** 3 good
**Contribution:** 1 poor

**Summary:**

This submission proposes 2 new geometric prior terms for fitting regular grids of SDFs to point clouds. The 2 priors are simulating the viscosity of the field to smooth it, and minimizing the area of the 0 level set.

**Questions:**

- How are gradients computed from the grid in eq. (3)?
- Why is Fig(2) showing a problem? the 0-isosurface is the same...
- In Tab. 2, why is Daratesh better without any additional loss term?
- 4.2 ablates different weights for the viscosity term. How hard is it to tune in practice?

**Limitations:**

A limitation not mentioned by the authors is that it is not possible to learn a shape prior with this formulation: only single shape fitting is possible.

**Strengths And Weaknesses:**

Strengths:
- technical soundness
- nice presentation with clear figures
- great ablation of the 2 added priors and their respective effects.

Weaknesses:
MAJOR - experiments are only conducted on 5 shapes only, no precise information is given about the optimization strategy (grid resolution, gradient descent steps...), and no normal consistency is provided.
MINOR - the introduction, the too short related work section, and the general scoping of the paper wrongly confuses 3D shape reconstruction from a point cloud with 3D shape representation with neural networks. Both are orthogonal research topics.
MINOR - the paper claims that using a grid of SDF has "clear benefits", but what are they? And the proposed 2 loss terms could be used on implciit networks as well..

---

> ### Author Response · Authors · 2022-08-02
> **Answer**
>
> We thank the reviewer for his detailed and thoughtful review. Below we address the reviewer’s questions and concerns.
>
> Q1: Experiments are only conducted on 5 shapes only.
>
> A1: The Surface Reconstruction Benchmark (SRB) [47] is a standard benchmark, and unfortunately other existing benchmarks lack the necessary licenses to use in a publication. We did however tested our method on a recently published benchmark of noisy real scans [22].
> We have used the exact same parameters from the SRB dataset (i.e., no special param tuning) and got top 1-3 results in all categories; In the revised paper, please see Table 2 and Figure 5 (in the new Sec. 4.1 “Surface Reconstruction from Real-Scans”).
>
> [22] Surface Reconstruction from Point Clouds: A Survey and a Benchmark, Zhangjin Huang, Yuxin Wen, Zihao Wang, Jinjuan Ren, Kui Jia; arXiv:2205.02413; 5 May 2022
>
> Q2: No precise information is given about the optimization strategy (grid resolution, gradient descent steps...), and no normal consistency is provided.
>
> A2: In the revised version of the supplementary we have added more details regarding optimization Sec. B, and timing/complexity Sec. C.  The new benchmark (see Table 2 in revised paper) has a normal consistency loss denoted NCS.
>
> Q3: The paper claims that using a grid of SDF has "clear benefits", but what are they? And the proposed 2 loss terms could be used on implicit networks as well.
>
> A3: The benefits of grid SDF representation (over neural network SDF representation)  are mostly: (i) controllable and better understood inductive bias, (ii) fast and easy inference; and (iii) faster training times. Regarding using our losses when training networks - possibly, but our main point is that these 2 losses on grids can be used *as a replacement* of the inductive bias injected by the networks.
>
> Q4: How are gradients computed from the grid in eq. (3)?
>
> A4: Please see Appendix A in the submitted (or the revised) supplementary.
>
> Q5: Why is Fig (2) showing a problem? the 0-isosurface is the same…
>
> A5: It can get arbitrarily close or actually cross 0, see for example Figure 1, middle column, where extraneous and noisy zero level-sets are added to the solution.
>
> Q6: In Tab. 2, why is Daratesh better without any additional loss term?
>
> A6: For this instance, although not having coarea loss achieves better quantitative performance, visual inspection reveals higher qualitative results for the mesh with coarea. We observe small holes when removing the coarea loss, see Figure II and Sec. D of the revised supplementary.
>
> Q7: 4.2 ablates different weights for the viscosity term. How hard is it to tune in practice?
>
> A7: Tuning the two hyper-param (Viscosity and Coarea) was rather easy. For example, in the new benchmark results we introduced in the revised version we didn't tune parameters at all.  In the revised paper supplementary Sec. B we have added a new paragraph dedicated to describe the full ranges of the Viscosity and Coarea hyper-params used.

---

> > ### Comment · Reviewer_tYGS · 2022-08-08
> > **Thank you for your answers**
> >
> > Thank you. I would advise in favor of moving the answers to Q2 and Q4 to the main paper.

---

> > > ### Author Response · Authors · 2022-08-08
> > > **Response**
> > >
> > > Thanks for the suggestion. We hope we addressed the reviewer’s main concern, as expressed in his review, with the addition of a new benchmark to the revised paper (see new Table 2 and new Figure 5).

---

### Official Review · Reviewer_5amW · 2022-07-15

**Rating:** 5
**Confidence:** 4
**Soundness:** 3 good
**Presentation:** 3 good
**Contribution:** 2 fair

**Summary:**

The paper proposes a new method for representing surfaces on grids using prior-based losses: viscosity loss and Coarea loss. They have shown better results than neural network SDFs in reconstructing surfaces on the Surface Reconstruction Benchmark.

**Questions:**

For the comparison, each sample is "optimized"/ "learned" on a separate grid? Does it downgrade to only solving a PDE, not a learning problem? It seems not so fair to compare with the neural network-based method, which can extend to "arbitrarily" resolution, while the grid is fixed.

**Limitations:**

The limitation mainly comes from the grid-based representation. It seems can only represent one model after optimization, unlike other network-based can use a latent code as an additional condition.

**Strengths And Weaknesses:**

Pro:

1. Instead of using a neural network as a bridge to build implicit surface representation, the authors propose to go back to using the grid-based method. It is easier to train than a neural network and is easier to find the zero surface.

2. The authors bring two new priors to the domain, viscosity and Coarea loss, to replace Eikonal loss and control the surface's area. They use the finite difference to compute the gradient or the Laplacian.

Con:

1. It does not directly compare how Viscosity loss and Coarea loss would perform with the neural network. The comparison with other works seems not fair.

---

> ### Author Response · Authors · 2022-08-02
> **Answer**
>
> We thank the reviewer for his detailed and thoughtful review. Below we address the reviewer’s questions and concerns.
>
> Q1: For the comparison, each sample is "optimized"/ "learned" on a separate grid? Does it downgrade to only solving a PDE, not a learning problem? It seems not so fair to compare with the neural network-based method, which can extend to "arbitrarily" resolution, while the grid is fixed.
>
> A1: Yes, we optimize each sample independently, and yes it amounts to solving a *non-linear* PDE. We are not sure we understand why comparison to Neural Network (NN)- based solution is not fair: NNs define a non-linear function space that was recently been advocated for using in an independent reconstruction scenario, which is the exact same scenario we consider, that is, train a different network for each sample.
>
> Q2: The comparison with other works seems not fair. The limitation mainly comes from the grid-based representation. It seems can only represent one model after optimization, unlike other network-based can use a latent code as an additional condition.
>
> A2: We are comparing our grid-based solution to Neural Network-based solutions in the *same setting*, namely training a grid/network on a single sample (i.e., an input point cloud) at a time.
>
> Q3: “It does not directly compare how Viscosity loss and Coarea loss would perform with the neural network”.
>
> A3: The goal of our paper is to enable effective surface reconstruction on grids by *replacing* the inductive biases of the neural networks with Viscosity and Coarea losses on grids. In that sense we claim the Viscosity loss and Coarea loss + grids should be used *as a replacement* to neural networks in surface reconstruction problems.

---

### Author Response · Authors · 2022-08-02
**General comment to all reviewers**

We thank the reviewers for their effort and detailed reviews. We responded to each reviewer individually addressing their questions and comments. We have uploaded a revised paper incorporating the reviewers’ requested changes (in blue). Notably, we added a new recently published benchmark of surface reconstruction from real scans where, without any further hyper-parameter tuning, our method achieved top 1 F-score and 2nd-3rd place in the other scores, compared to a wide range of baselines. Please note: The revised paper also includes a revised supplementary concatenated at the end.

---

### Meta-Review · Area_Chair_WQbD · 2022-08-23

**Recommendation:** Accept
**Confidence:** Certain

**Metareview:**

The paper originally received mixed scores, with two reviewers recommending acceptance and two rejection. While the reviewers acknowledged the soundness of the approach, the clarity of the explanations, and the good ablations of the method's components, they expressed concerns regarding the limited scope of the experiments, the lack of runtime analysis, and the fairness of the comparison to baselines. The authors' feedback convincingly addressed these concerns; in the discussion, tYGS agreed to raise their score to Weak Accept, and ubX9 to borderline accept. This led to a consensus for acceptance. We nonetheless strongly encourage the authors to incorporate elements of their feedback in the camera-ready version of their paper.

**Award:**

No

---

### Decision · Program_Chairs · 2022-09-14

Accept